# A CONFORMALIZED INFERENCE ON UNOBSERVABLE VARIABLES

## ABSTRACT

Quantifying uncertainty in predicted unobservable variables is a critical area of research in statistics, artificial intelligence, and empirical science. Most scientific studies assume a specific structure involving unobservable variables for the data-generating process and draw inferences from a parameter of interest within that framework. Conformal prediction is a popular model-agnostic method for constructing prediction intervals for new observations. However, it typically requires observed true labels to build the prediction interval, making it unsuitable for unobserved latent variables. We propose a method to construct a prediction interval by leveraging sample-splitting of the training data and analyzing the discrepancy between two independently trained models. To ensure the identifiability of the distribution of this conformity score, we introduce a few assumptions regarding the distribution of the residuals of the predictions. Furthermore, we propose a residual orthogonalization to satisfy these assumptions with a coordinating regularization term. The performance of the proposed method was evaluated using both simulation and large language model experiments.

## 1 INTRODUCTION

Scientific inquiry frequently necessitates inferring unobservable variables from observed data to address quantitative research questions. These latent variables, despite their indirect nature, often play a crucial role in decision-making across diverse domains. In clinical diagnostics, for instance, physicians must deduce underlying pathologies from a limited set of observable symptoms and test results (Ledley and Lusted, 1959; Kassirer, 1989). Similarly, in clinical research, the estimation of treatment effects relies on incomplete observational data, requiring sophisticated inference about counterfactual outcomes (Pearl, 2009; van der Laan and Rose, 2011; 2018; Hernan and Robins, 2024). The significance of these unobserved variables in practical decision-making underscores the importance of robust uncertainty quantification methods, such as prediction intervals. These measures provide decision-makers with critical insights into the reliability and variability of inferred information, thereby enhancing the quality of decisions in fields ranging from healthcare to policy formulation. Consequently, the development and application of advanced inferential techniques remain central to progress in both theoretical and applied sciences.

Many modern applications rely on inferring latent variables from observed data using complex models such as deep neural networks. In LLM alignment (Ouyang et al., 2022), for instance, the goal is to infer an implicit reward signal—unobserved human preferences—from pairwise comparison data, and use it to fine-tune language models. This reward modeling process is inherently uncertain, and recent work suggests that incorporating uncertainty improves robustness and alignment performance (Laidlaw and Russell, 2021; Zhai et al., 2023; Zhang et al., 2024). In education research, LLMs were used to generate questions with students' understanding taken into account based on the estimate using the item response theory (Srivastava and Goodman, 2021; Uto et al., 2023). Longitudinal treatment effects given the patients' health trajectory can be estimated as a value function in a non-Markovian decision process with deep neural networks (Frauen et al., 2023; Shirakawa et al., 2024; Hess et al., 2024). Deep neural networks were also used for survival analysis that estimates the hazard functions (Katzman et al., 2018; Steinberg et al., 2024).

Conformal prediction (CP) (Vovk et al., 2005) is a model agnostic method to quantify the uncertainty of the out-of-sample prediction by constructing prediction intervals. However, CP may be less useful

for classification tasks with a small number of classes because the constructed prediction set tends to include all possible class labels. For example, when CP is applied to a binary classification task, the resulting 90% prediction set tends to contain both labels if the model has high uncertainty, leading to an uninformative confidence set. To illustrate the challenge, we begin with binary classification. However, the goal of this work is broader; we develop a general method for constructing prediction intervals for unobserved latent variables, of which classification probabilities are just one example. For instance, we construct a prediction interval of the difference of the implicit rewards of the Bradley-Terry model (Bradley and Terry, 1952), then applied it to preference learning for fine-tuning the large language models (LLMs).

The key contributions of this study are:

- We develop a Latent Conformal Prediction (LCP) method to construct prediction intervals for unobservable latent variables. The key idea is to estimate the distribution of prediction residuals without access to ground-truth labels, by leveraging sample-splitting and analyzing the discrepancy between two independently trained models. This allows us to extend conformal prediction to settings where the variable of interest is never observed, such as in preference learning or probabilistic classification.

- We further introduce a procedure, called residual orthogonalization, for finding an estimator that meets the assumptions required to identify the distribution of the conformity score. For this purpose we introduced a regularization term in a coordinated manner.

- The performance of the proposed algorithm is evaluated with synthetic data of binary classification and preference learning for LLMs. We demonstrated the reasonable interpretation of the proposed method with a few examples from a human preference data over paired model-generated responses.

## 2 RELATED WORK

CP is a method for producing prediction sets that guarantees coverage of the true label (Vovk et al., 2005). The prediction sets are constructed by taking a quantile of conformity score, which measures how close the prediction is to the true label, computed in a randomly separated calibration set from the training set on which the prediction model is trained. A common choice of the conformity score is the distance between the true label and the prediction. CP for regression problems, where the true label is usually accessible, has been an area of extensive research (Vovk et al., 2005; Gammerman et al., 1998; Lei et al., 2013; 2018; Alaa et al., 2023).

CP for binary and multinomial classification have been proposed (Sadinle et al., 2019; Romano et al., 2020; Angelopoulos et al., 2021; Ding et al., 2023; Huang et al., 2024). Guha et al. (Guha et al., 2024) applied this classification CP to regression problem to construct prediction sets for heteroscedastic, multimodal, or skewed distributions. Applications of classification CP in medicine were also published for predicting risks of breast cancer (Lambrou et al., 2009) and stroke (Lambrou et al., 2010). However, all these studies output discrete prediction sets of classification labels. Although the conformity score for the binary classification probability can be measured by the distance between the binary label and the predicted probability, this score would be too large to be of practical use.

A method for distribution-free binary classification with prediction intervals for classification probabilities was proposed by Gupta et al. (2020). Their method assumes that the latent variable is bounded and can be expressed as a conditional expectation of some observable variable, which enables the use of concentration inequalities such as Bernstein's. However, this formulation does not apply to preference learning, where the latent reward is typically unbounded and cannot be represented as an expectation of any observable variable, even after transformations such as the sigmoid function. Feature conformal prediction (Teng et al., 2023; Chen et al., 2024) is a method to construct efficient, i.e., narrower, prediction intervals using a latent feature of an observable continuous outcome. However, their method could only be applicable to the continuous $Y$, and cannot be applied to our problem of binary classification or preference learning where we observe discrete labels.

Motivated by this gap between CP for continuous regression and discrete classification, we explore the situation where the latent variable of (transformed) probability can be identified from the observed

labels, hence the standard techniques of CP are applicable. The problem can be stated formally in a general setting of CP for latent variables.

## 3 PRELIMINARIES

In this section, we provide the foundational concepts necessary for our proposed method. We begin by introducing conformal prediction, a model-agnostic framework for uncertainty quantification, which serves as the basis for our approach (Section 3.1). We then discuss latent variables, which play a central role in many statistical models, highlighting the challenges and importance of uncertainty estimation in this context (Section 3.2).

### 3.1 CONFORMAL PREDICTION

Conformal prediction constitutes a methodological framework designed to quantify the uncertainty associated with predictions in machine learning. This framework provides a measure of confidence by generating prediction intervals that are expected to contain the true value with a specified probability. We provide a concise exposition of the standard conformal prediction algorithm for regression tasks.

Suppose that a dataset comprises $n$ observations $D := \{(X_i, Y_i)\}_{i=1}^n$, where $X_i \in \mathcal{X} \subseteq \mathbb{R}^{d_X}$ and $Y_i \in \mathcal{Y} \subseteq \mathbb{R}$. The objective of conformal prediction is to estimate the prediction interval $\hat{C}_{1-\alpha}(X_{n+1})$ for $\alpha \in (0, 1)$ such that, given a new sample $(X_{n+1}, Y_{n+1})$,

$$\Pr[Y_{n+1} \in \hat{C}_{1-\alpha}(X_{n+1})] \geq 1 - \alpha. \tag{1}$$

To estimate this interval, the standard conformal prediction algorithm partitions the data $D$ into a training set $D^{\mathrm{train}}$ and a calibration set $D^{\mathrm{cal}}$. Subsequently, we train a prediction model $f : \mathcal{X} \to \mathcal{Y}$ using the training data $D^{\mathrm{train}}$ and we use the conformity scores computed from the trained model $\hat{f}$. One of commonly employed conformity scores in regression is the absolute residual:

$$|Y_i - \hat{f}(X_i)|, \tag{2}$$

for $(X_i, Y_i) \in D^{\mathrm{cal}}$. The conformity scores are then sorted to obtain the $(1 - \alpha)(1 + |D^{\mathrm{cal}}|^{-1})$-quantile, denoted as $\hat{Q}_{1-\alpha}$. Finally, the prediction interval is estimated as

$$\hat{C}_{1-\alpha}(X_{n+1}) = \left[\hat{f}(X_{n+1}) - \hat{Q}_{1-\alpha}, \hat{f}(X_{n+1}) + \hat{Q}_{1-\alpha}\right].$$

It has been demonstrated that this interval satisfies the inequality equation 1.

### 3.2 LATENT VARIABLES

Let $Y \in \mathcal{Y}$ denote the response variable and $X \in \mathcal{X}$ denote the predictor variables. Many statistical models incorporate latent variables $Z$, which represent unobservable factors that influence the relationship between predictors and response. We consider latent variable models expressed by the following structural model:

$$X \longrightarrow Z \longrightarrow Y,$$

where the response $Y$ given the observed data $X$ is modeled through a latent variable $Z$: $\Pr(Y|X) = \Pr(Y|Z)$ with $Z$ produced from $X$ by some function $f$ as $Z = f(X)$. Here, we assume $f$ is deterministic, which holds in many practical AI applications, including logits in classification problems, rewards in RLHF (Ouyang et al., 2022), item parameters in item response theory (Hambleton and Swaminathan, 1985), and conditional intensities in survival analysis (Andersen et al., 1996).

In many cases, latent variables are interpretable and can guide crucial decisions, such as treatment choices in healthcare (Muthén, 1992) or strategic decisions in marketing (Draganska et al., 2008). Accurately quantifying uncertainty, i.e., constructing prediction intervals for estimated latent variables, is crucial for these practical applications. The ability to determine precise interval lengths directly impacts the reliability of decisions made based on the latent variable estimates.

Estimating such prediction intervals poses significant challenges. Approaches like Bayesian inference and bootstrapping are commonly used to quantify uncertainty in latent variable estimates. However, Bayesian inference depends heavily on the choice of prior distributions, which can influence the resulting coverage rates and make it difficult to ensure that intervals capture the true latent variable with a specified probability (e.g., 90%).

## 4 PROBLEM AND APPLICATIONS

In this section, we first formulate the problem we address in the present study. Then, we will proceed with practical applications of the binary classification, and the preference learning and its multinomial extensions. Our method can be easily extended to more general cases of latent variable modeling.

### 4.1 PROBLEM

Our objective is to construct prediction intervals for the latent variable $Z_{n+1}$ corresponding to a new observation $X_{n+1}$, such that these intervals achieve a pre-specified target coverage probability.

$$\Pr[Z_{n+1} \in \hat{C}_{1-\alpha}(X_{n+1})] \geq 1 - \alpha, \tag{3}$$

given a dataset $\{(X_i, Y_i)\}_{i=1}^n$. The fundamental challenge arises from the absence of direct observations of $\{Z_i\}_{i=1}^n$ in our dataset. This precludes the application of conventional methodologies that utilize prediction error metrics such as $|Z_i - f(X_i)|$ for uncertainty calibration as in equation 2.

### 4.2 APPLICATION 1: BINARY CLASSIFICATION

Let $Y \in \{0, 1\}$ be a binary label and $X \in \mathbb{R}^{d_X}$ be a feature random vector, where $d_X$ is the dimension of $X$. Suppose that $Y$ is generated from a Bernoulli distribution with success probability $\sigma(Z)$ for a scalar latent variable $Z$, where $\sigma(z) := (1 + e^{-z})^{-1}$ is the sigmoid function. In this setting, the latent variable of interest is $Z = \text{logit}(\mathbb{E}[Y|X])$, where $\text{logit}(\cdot) = \sigma^{-1}(\cdot)$. Let $\{(X_i, Y_i)\}_{i=1}^n$ be the samples for training, where $n$ is the sample size, and $X_{n+1}$ the out-of-sample feature for which we want to construct the prediction interval $\hat{C}_{1-\alpha}(X_{n+1})$ for $Z_{n+1}$ satisfying the inequality equation 3.

When we consider this problem as a prediction problem of $Y$ based on $X$ with the target function $\mu(x) = \text{logit}(\mathbb{E}[Y|X = x])$, then CP typically uses the absolute residuals $|Y_i - \sigma(\mu(X_i))|$ as the conformity score as described in Section 3.1 with a rescale of the $(1 - \alpha)$-quantile $Q$ by the logit function. However, in the classification task, the label is binary and thus the computed prediction interval might be too conservative (wide), potentially exceeding the valid probability range of zero to one. Therefore, we formulate the problem as the estimation of the logit of the probability and estimate its prediction interval. The difficulty here is that we do not have access to the true values of the logit in the observations.

### 4.3 APPLICATION 2: PREFERENCE LEARNING

Preference learning from real-world data is an important area of machine learning and artificial intelligence with a wide range of applications including recommendation system, reinforcement learning from human feedback, and clinical decision support for evidence-based medicine (Tsopra et al., 2018). The Bradley-Terry (BT) model (Bradley and Terry, 1952) is a popular method for preference learning assuming the probability of preference is determined by the difference of implicit rewards for two items compared. The BT model has been used to fine-tune large language models by aligning them with human preferences (Ouyang et al., 2022; Touvron et al., 2023). The BT model was extended to ranking learning by the softmax function of multiple implicit rewards of candidate items (Luce, 1959; Plackett, 1975).

**Bradley-Terry model.** Preference learning task is formulated with the observed data $(W, Y_0, Y_1, L)$, which consists of independent identically distributed random variables. Here, $W$ represents user characteristics, while $Y_0$ and $Y_1$ denote the items being compared. The binary variable $L \in \{0, 1\}$ indicates the chosen item, where $L = 1$ if $Y_1$ was selected and $L = 0$ if $Y_0$ was selected. We introduce the Bradley-Terry model:

$$\Pr(Y_1 \succ Y_0 \mid W) = \sigma\big(r(W, Y_1) - r(W, Y_0)\big), \tag{4}$$

where $r$ is the implicit reward function. Defining the binary indicator $Y$ as $Y = L$, we can interpret the preference learning task as a binary classification problem. Instead of the logit of the probability of $Y = 1$, the problem is not normalized because there is one dimensional freedom for translations of the reward $r(W, Y)$. After applying a normalization, for example, $r(W, 0) = 0$ or $r(W, 0) + r(W, 1) = 0$, we can identify the reward function. Herein, we also face a challenge of constructing prediction intervals of individual preference, or reward, using CP because we cannot directly observe these implicit rewards in data.

---

**Algorithm 1:** Latent Conformal Prediction (LCP)

---

$1 : D \to D^1 \cup D^2 \cup D^{\mathrm{cal}}$                                `// sample split;`

$2 : \hat{f}^j \leftarrow$ train the model with $D^j$ for $j = 1, 2$;

$3 : \tilde{f} \leftarrow (\hat{f}^1 + \hat{f}^2)/2$;                                   `// final predictor;`

$4 : \tilde{U}_k \leftarrow (\hat{f}^1(X_k) - \hat{f}^2(X_k))/2$ for $k \in D^{\mathrm{cal}}$;

$5 : Q \leftarrow (1 - \alpha)(1 + n^{-1})$-quantile of $\{|\tilde{U}_k|\}_{k \in D^{\mathrm{cal}}}$;

$6 : \hat{C}(X_{n+1}) \leftarrow [\tilde{f}(X_{n+1}) - Q, \tilde{f}(X_{n+1}) + Q]$;         `// prediction interval`

---

## 5    LATENT CONFORMAL PREDICTION

We propose a method for conformal inference on latent variables which consists of 1) a reduction of the problem to the standard conformal prediction with assumptions, and 2) an algorithm to make the estimator satisfy the assumptions. Specifically, we first derive a model-agnostic method for constructing a prediction set based on assumptions on residuals of the first-stage estimators. Second, we propose a method to have the estimators meet the criteria assumed in the first part. Then, we extend the method for the localized prediction set construction.

### 5.1    ALGORITHM

Since the true latent value $Z$ is unobservable, we cannot directly compute the residual error

$$\tilde{U} := \tilde{f}(X) - Z, \tag{5}$$

for some prediction model $\tilde{f}$. Instead, we introduce a proxy variable for computing the distribution of the residual $\tilde{U}$. We propose a latent conformal prediction (LCP) that uses a sample splitting of dataset into three independent subsets:

$$D = D^1 \cup D^2 \cup D^{\mathrm{cal}}. \tag{6}$$

The datasets $D^1$ and $D^2$ are used to train the prediction models $\hat{f}^1$ and $\hat{f}^2$ for the latent variables separately. We will prove in Proposition 2 that under some assumptions the distribution of $\tilde{U}$ for the averaged prediction model $\tilde{f} := (\hat{f}^1 + \hat{f}^2)/2$ can be identified via the proxy variable defined by

$$\hat{V} := \hat{f}^1(X) - \hat{f}^2(X), \tag{7}$$

on the calibration set $D^{\mathrm{cal}}$. Note that the proxy variable can be expressed as the difference between the two residual errors $\hat{V} = \hat{U}^1 - \hat{U}^2$, where $\hat{U}^j = \hat{f}^j(X) - Z$. Note that $\hat{U}^j$ cannot be predicted from observed data since they involve the unobserved quantity $Z$.

The procedure is illustrated in Algorithm 1, whose coverage equation 3 is theoretically guaranteed by Theorem 1 under Assumptions 1, 2 and 3 introduced in the next subsection. The following subsection presents a formal theory behind this algorithm, followed by an approach to ensure that the assumptions hold by appropriately training the models with the sample split equation 6.

### 5.2    THEORY

In this subsection, we describe the identification results for the distribution of residuals under a few assumptions. This identification requires building estimators that are "independent" in a certain sense, which may sacrifice statistical efficiency. We will also introduce a method to recover this efficiency in the subsequent arguments. This method for efficiency recovery proves useful in identifying the distribution of the conformity score.

To identify the distributions of $\hat{U}^1$ and $\hat{U}^2$, we use a result from the measurement error literature. In that field, the problem is formulated as two measurements $Z^j = Z + U^j$, where $Z$ is the variable of interest and $U^j$ are measurement errors. Under the assumptions of independence of $Z$ and $U^j$, one can identify the distribution of $Z$ and $U^j$. However, our interest here is the distribution of residuals $U^j$. Given this context, we assume the following conditions on the residuals.

**Assumption 1** (Orthogonality). *The residuals are independent: $\hat{U}^1 \perp \hat{U}^2$.*

**Assumption 2** (Identity). *The residuals have the identical distribution: $\hat{U}^1 \sim \hat{U}^2$.*

**Assumption 3** (Symmetry). *The residuals are symmetric: $\hat{U}^j \sim -\hat{U}^j$ for $j = 1, 2$.*

**Remark** (Role of Assumption 2). Assumption 2 (identity of residual distributions) is included to present the classical form of Kotlarski's lemma (Proposition 1), which many readers may find familiar from the measurement-error literature. However, and this is an important point, our main identifiability result (Proposition 2) and the coverage guarantee (Theorem 1) do NOT rely on Assumption 2. They only require Assumptions 1 and 3. We therefore keep Assumption 2 for consistency with prior work and for ease of mapping to reviewer feedback, but it plays no essential role in the main theory developed in this paper.

Under Assumptions 1, 2 and 3, the distribution of the error, denoted simply as $\hat{U}$ since $\hat{U}^1 \sim \hat{U}^2$ by Assumption 2, is identified by a simplified Kotlarski's Lemma (Kotlarski, 1967; Li and Vuong, 1998):

**Proposition 1** (Kotlarski). *Under Assumptions 1, 2 and 3, we can identify the distribution of $\hat{U}$ via the proxy $\hat{V}$ as*

$$\varphi_{\hat{U}} = \sqrt{\varphi_{\hat{V}}}, \tag{8}$$

*where $\varphi_X(t) = \mathbb{E}[e^{iXt}]$ is the characteristic function of a random variable $X$.*

**Remark.** In general, Assumption 1 does not hold, especially when employing small models. The independence between $\hat{U}^1$ and $\hat{U}^2$ is a conditional independence given a sample split of $D^1$ and $D^2$. In the next section we will describe a technique to satisfy this assumption by trying different splitting with hypothesis tests. On the other hand, Assumption 3 would be more reasonable when we carefully choose the model, or the loss function. For example, in classification or preference learning tasks, the logits of the probability are of our interest. The common choice of the loss function is a cross entropy loss:

$$\mathcal{L}(\hat{y}, y) = \sum_{i=1}^{K} y_i \log \hat{y}_i,$$

which is linear in the logits of $\hat{z}_i = \log p_i$, where $y_i \in \{0, 1\}$ is the binary classification label and $\hat{z}_i$ is the logit of the probability of being classified as the class $i$ for $i = 1, \ldots, K$, where $K$ is the number of classes. Therefore, we could expect the symmetric fluctuation of the predictions with respect to the random perturbation of the observation in the training set.

**Remark.** Note that the characteristic function $\varphi_X$ of a random variable $X$ is the Fourier transform of the density function of $X$. The density of $X$ is recovered by the inverse Fourier transform from the characteristic function. Therefore, the distribution of $\hat{U}$ is identified through the distribution of $\hat{V}$ by Proposition 1. Numerical computation based on the identification equation 8 requires two steps of the Fourier transform and the inverse Fourier transform after taking the square root of the characteristic function. Previous econometrics studies of measurement error proposed a method using smoothing kernels because the first transform involves a Fourier transform of empirical distribution which needs to be smoothed out to recover the distribution (Li and Vuong, 1998; Kurisu and Otsu, 2022). Furthermore, the discretization levels and the range of integration of Fourier transform is to be determined by the user since the characteristics function of the empirical distribution does not have a compact support. However, we could solve this problem by avoiding the Fourier transform simultaneously gaining efficiency in an additional identification result stated in Proposition 2.

**Corollary 1.** *For a Gaussian $\hat{U}$, we have*

$$\hat{U} \sim \hat{V}/\sqrt{2}. \tag{9}$$

**Remark.** The relationship $\hat{U} \sim \hat{V}/\sqrt{2}$ in Corollary 1 greatly simplifies the numerical computation of the distribution of conformity score because it does not require Fourier transforms, and thus improves the quality of estimates. However, this does not hold for general distributions even for those satisfying Assumptions 1, 2 and 3. For example, the characteristics function of the standard Laplace distribution is $\varphi(t) = 1/(1 + t^2)$ and $\varphi^2(t)$ is no longer the characteristics function of a scaled Laplace distribution by a factor of $1/\sqrt{2}$. We proceed further to establish the identifiability of the distribution of $\hat{U}$ from residuals under substantially weaker conditions.

**Efficiency Recovery.** We can construct a more efficient estimator by averaging the two estimators with its conformity score identified by a very simple form:

**Proposition 2.** *Under the Assumptions 1 and 3, we can identify the distribution of the residual $\tilde{U}$ as*

$$\tilde{U} \sim \hat{V}/2. \tag{10}$$

**Remark.** Note that the identification result $\tilde{U} \sim \hat{V}/2$ in Proposition 2 holds for any distributions satisfying Assumptions 1 and 3 not limited to Gaussian distributions as in Corollary 1 and the following remarks. By averaging the models trained in the split dataset, we not only recover the efficiency in the predictive ability but also identify the distribution of the conformity score $\tilde{U}$ only with the nonparametric assumptions that immediately yields the equivalence: $\hat{U}^1 + \hat{U}^2 \sim \hat{U}^1 - \hat{U}^2$.

**Theorem 1.** *Under Assumptions 1 and 3, the prediction interval constructed in Algorithm 1 satisfies the coverage:*

$$\Pr[Z_{n+1} \in \hat{C}_{1-\alpha}(X_{n+1})] \geq 1 - \alpha. \tag{11}$$

## 5.3 RESIDUAL ORTHOGONALIZATION

Since Assumption 1 does not hold generally, we propose a remedy to satisfy the assumption through regularization. We introduce an additional simultaneous optimization of two models after the first step of Algorithm 1 with a regularization term to make the residuals of the first and the second models independent which are calculated using the third model $f_{\theta_3}$ fitted on $D^{\text{cal}}$:

$$\mathcal{L}(\theta_1, \theta_2; \theta_3) = \mathcal{L}_1(\theta_1) + \mathcal{L}_2(\theta_2) + \gamma \mathcal{L}^{\perp}(\theta_1, \theta_2; \theta_3), \tag{12}$$

where $\mathcal{L}_j$ are the loss functions of the models $f_{\theta_j}$ computed with samples from each split $j \in \{1, 2\}$, and $\mathcal{L}^{\perp}$ is the orthogonal loss defined as:

$$\mathcal{L}^{\perp}(\theta_1, \theta_2; \theta_3) = \frac{1}{B} \sum_{i=1}^{B} U_i'(\theta_1; \theta_3) U_i'(\theta_2; \theta_3), \tag{13}$$

where $B$ is the batch size, and $U_i'(\theta_j; \theta_3) = \hat{Z}_i(\theta_j) - \hat{Z}_i(\theta_3)$ with $\hat{Z}_i(\theta_j) = f_{\theta_j}(X_i)$ for each split data $j \in \{1, 2, 3\}$ and individual $i$ in the calibration set. We optimize the loss equation 12 with respect to $\theta_1$ and $\theta_2$ to ensure the residuals are independent, as required by Assumption 1.

### 5.3.1 THEORETICAL JUSTIFICATION

Here we describe the theoretical justification of the residual orthogonalization with the loss 13. Let $\text{HSIC}(X, Y)$ be the Hilbert-Schmidt Independence Criteria (HSIC) (Gretton et al., 2007) of random variables $X$ and $Y$.

**Theorem 2.** *Suppose the kernel for the HSIC is bounded and 1-Lipchitz. Then the true HSIC is bounded by its proxy and terms in the residual $\Delta := \hat{Z}(\theta_3) - Z$ up to a universal constant:*

$$\text{HSIC}(U(\theta_1), U(\theta_2)) \lesssim \text{HSIC}(U'(\theta_1; \theta_3), U'(\theta_2; \theta_3)) + \mathbb{E}[|\Delta|] + \mathbb{E}[\Delta^2]. \tag{14}$$

The orthogonal loss $\mathcal{L}^{\perp}$ is a special case of HSIC with a linear kernel $k(u, v) = uv$. Hence our algorithm corresponds to the first-order orthogonalization of the residuals. The orthogonal loss equation 13 corresponds to the empirical mean of the first term of the right hand side of equation 18 with this first order kernel. Therefore, the optimization of equation 13 guarantees the optimization of the HSIC of the true residuals up to the error terms involving $\Delta$.

## 5.4 LOCALIZED LATENT CONFORMAL PREDICTION

We further explore the localized conformal prediction that provides the adaptive confidence interval $\hat{C}_{1-\alpha}(X_{n+1})$ depending on the value of the feature value:

$$\Pr[Z_{n+1} \in \hat{C}_{1-\alpha}(X_{n+1})|X_{n+1}] \geq 1 - \alpha. \tag{15}$$

To construct such intervals, we assume a scalar localization score $\Gamma(x) \in \mathbb{R}$ such that the conditional distribution of $\tilde{U}$ varies primarily through $\Gamma(x)$. Intuitively, this induces a one-dimensional stratification under which the conditional dispersion of $\tilde{U}$ is approximately stable within small neighborhoods

of $\Gamma(x)$. In our implementation, $\Gamma(x)$ is instantiated as a density-based score of $X$, which typically induces wider intervals in low-density regions and narrower ones in high-density regions.

Let $K$ be a positive integer and let $\mathcal{U}(x)$ denote the $K$ calibration points whose scores $\Gamma(X_i)$ are closest to $\Gamma(x)$ in the calibration set $D^{\mathrm{cal}}$. We compute the local quantile $Q_{1-\alpha}(x)$ as the $(1-\alpha)(1+|n|^{-1})$-th quantile of $\mathcal{V}(x) = \{\tilde{U}_i \mid i \in \mathcal{U}(x)\}$, and then construct the prediction interval as

$$\hat{C}_{1-\alpha}(x) = [\tilde{f}(x) - Q_{1-\alpha}(x), \tilde{f}(x) + Q_{1-\alpha}(x)]. \tag{16}$$

We call this approach as localized latent conformal prediction (localized LCP).

## 6 EXPERIMENTS

We conducted experiments to evaluate the proposed method in two settings: a synthetic binary classification task and an LLM-based preference ranking task. See Appendix B for the detailed experiment settings. We also evaluate the utility of these intervals in LLM alignment via RLHF; see Appendix C.2 for details and results.

### 6.1 SIMULATION EXPERIMENT

We considered a simple simulation setup: $X \sim \mathcal{N}(0, I^3)$, $Z = 3\sin(\bar{X})$, and $Y \sim \mathrm{Ber}(\sigma(Z))$. We generated 3000 samples, split them into $D^1$, $D^2$, and $D^{\mathrm{cal}}$ (1000 each), and included a 90% bootstrap baseline (500 resamples) and the method proposed by Gupta et al. (2020). See Appendix B.1 for full details and Appendix B.2 for additional results for the BT model.

The results are shown in Figure 1. The proposed method showed better coverages with larger lengths as the hyperparameter $\gamma$ increases than bootstrap. MSEs of the proposed method were comparable with that of bootstrap. As the hyperparameter $\gamma$ increases, the MSE slightly decreased, suggesting that the regularization term in the loss equation 12 contributed to better predictive performance.

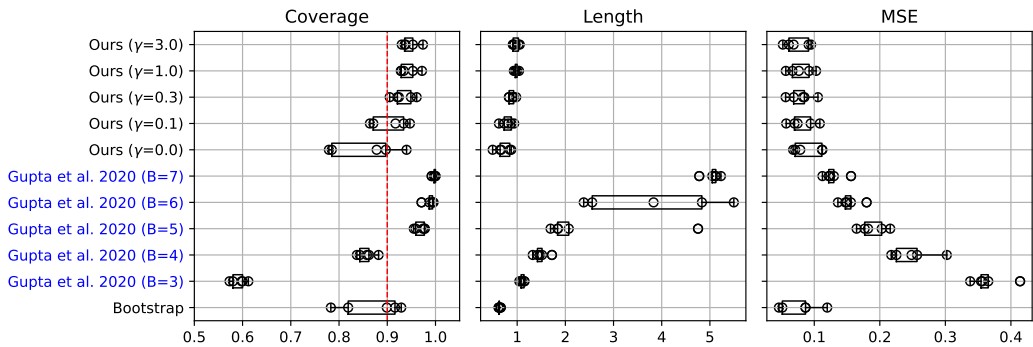

Figure 1: Performance comparison with synthetic data. "Length" refers to the mean interval length for the model by Gupta et al. (2020).

### 6.2 LLM PREFERENCE RANKING EXPERIMENT

As a more realistic example, we evaluated the method on an LLM preference ranking task using the AlpacaFarm dataset (Dubois et al., 2023). We fine-tuned the pretrained LLM-based reward model `Skywork-Reward-Llama-3.1-8B-v0.2` (Liu et al., 2024), where $X$ denotes the 4096-dimensional final-layer representation of the backbone LLM that feeds the reward head. We used `URM-LLaMa-3.1-8B` (Lou et al., 2024) as the gold reward model both to provide ground-truth preference labels (binary indicators of the preferred response in a pair) for evaluation and to generate training preference labels via a BT model. We split the dataset into $D^1$, $D^2$, $D^{\mathrm{cal}}$, and $D^{\mathrm{eval}}$ (2400 samples each). See Appendix C.1 for additional dataset and training details.

The results are shown in Figure 2. The proposed method approached the target coverage as $\alpha$ increased, while naive LCP ($\alpha = 0$) underperformed, likely due to correlations between residuals without the regularization term in equation 12.

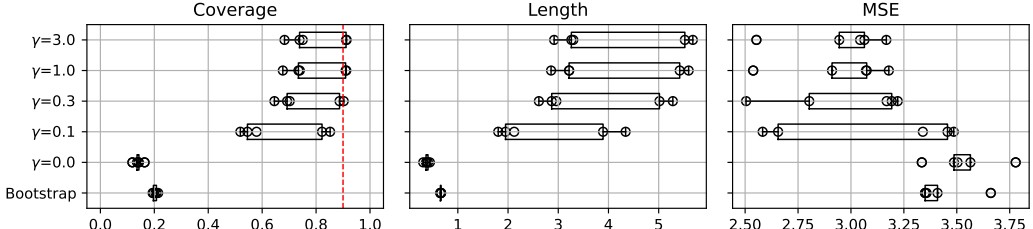

Figure 2: Performances of the proposed methods compared with the bootstrap with the pretrained reward model `URM-LLaMa-3.1-8B`.

## 6.3 ADAPTIVE INTERVAL PREDICTION EXPERIMENT

We conducted experiments using the LLM datasets and models described in Section 6.2 to evaluate the effectiveness of localized LCP. To construct a localized prediction interval, we first computed a density-based localization score $\Gamma(x)$ for each input $x$.

We performed principal component analysis (PCA) on $X$ in the training datasets $D^1$ and $D^2$, retaining the top 100 components to obtain a low-dimensional representation. On this reduced space, we fitted a kernel density estimator $\hat{p}$ using the training datasets $D^1$ and $D^2$. We defined the localization score as $\Gamma(x) = \hat{p}(z(x))$, where $z(x)$ denotes the PCA representation of $x$. Given a new input $x$, we selected the $K = 500$ calibration points $x' \in D^{\text{cal}}$ with scores $\Gamma(x')$ closest to $\Gamma(x)$ and used their conformity scores $\tilde{U}_{x'}$ to compute the local quantile $Q_{1-\alpha}(x)$. The prediction interval was then constructed as in equation 16.

To evaluate empirical conditional (local) coverage, we defined, for each point $x$ in the evaluation set, a neighborhood $\mathcal{U}(x)$ in the localization score space via $\Gamma(x)$. We then measured the proportion of points whose latent variable $Z$ lies within the corresponding interval $\hat{C}_{1-\alpha}(x)$.

The results are shown in Figure 3: standard LCP suffers from over-coverage in low-density regions and under-coverage in high-density ones due to its constant interval length, whereas localized LCP adjusts the interval width according to the localization score $\Gamma(x)$ and yields better-calibrated local coverage across the feature space.

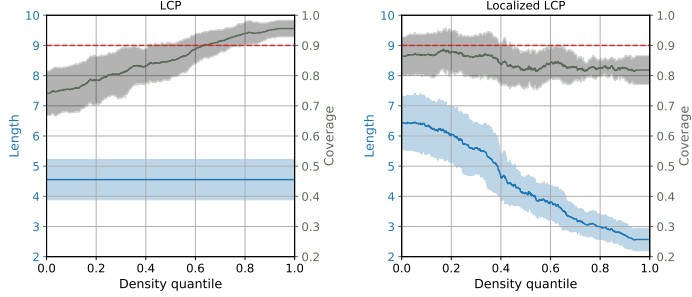

Figure 3: Comparison of standard LCP (left) and localized LCP (right) across density quantiles (i.e., quantiles of the density-based localization score $\Gamma(x)$). Localized LCP achieves adaptive interval lengths and better-calibrated local coverage.

## 7 CONCLUSION

This study develops a method to construct prediction intervals for unobserved latent variables by estimating conformity scores through sample splitting. Our theory establishes identifiability of the residual distribution and coverage guarantees for the resulting intervals. We demonstrated the utility of the proposed method in a synthetic binary classification task and an LLM preference learning task.

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

# A    BINARY CLASSIFICATION AS REGRESSION PROBLEM

Here we provide a detailed exposition of binary classification task as conformal prediction (CP) for regression, which we briefly mentioned in Section 4.1 in the manuscript. The interval prediction $\hat{C}_{1-\alpha}(X_{n+1})$ of $Y$ can be constructed from the interval prediction $\hat{C}_{1-\alpha}(X_{n+1}; \hat{f})$ of $f(x) = \mathbb{E}[Y|X = x]$ by the method described in Section 3.1 with a conformity score $|Y - \hat{f}(X)|$. One might consider using this interval to cover the true latent variable $Z = \mathbb{E}[Y|X]$. We computed the coverage and length of the prediction interval by this native CP with a synthetic data described in Section 6.1. The average length of these intervals were $1.296 \pm 0.018$ with a mean coverage of the observable label $Y_{n+1}$ of 90.0% as expected. However, the coverage of the true latent variable $Z_{n+1}$ by this intervals was 100% in all 100 runs of experiments, implying that the prediction interval for the regression of $Y$ is too conservative for the latent variable $Z$ in practice. The oracle coverage computed with an oracle conformity score $|Z - \hat{f}(X)|$ for predicting $Z$ was 90.0% with a mean length of $0.189 \pm 0.019$.

# B    EXPERIMENTS DETAILS

## B.1    SIMULATION EXPERIMENTAL SETUP

We used a simple parametric model to generate a synthetic data as $X \sim \mathcal{N}(0, I^3)$, $Z = 3\sin(\bar{X})$, and $Y \sim \mathrm{Ber}(\sigma(Z))$. We generated 3000 samples based on this data generating process, and split this dataset into two sets of training sets, $D^1$ and $D^2$, and a calibration set with 1000 samples each. The performance metrics of the coverage $n^{-1}\mathbb{1}\{Z_i \in \hat{C}(X_i)\}$, the length $2Q$ of the prediction set, and the MSE $n^{-1}\sum_{i=1}^{n}\left[\hat{f}(X_i) - Z_i\right]^2$ are calculated with another independent dataset $D^{\mathrm{eval}}$, consisting of 1000 samples.

We trained a neural network with two dense linear layers of dimension 64, drop-out layers after each layer with a drop-out rate of 0.1, followed by the rectified linear unit (ReLU) activate functions to learn the logits. We trained the model using the Adam optimizer (Kingma and Ba, 2015) with a learning rate of 0.001 and a batch size of 128 through 50 epochs.

For the baseline model, we implemented the method proposed by Gupta et al. (2020). Their method is based on the binning of the feature space $\mathcal{X}$ according to the predicted values $f(X)$ given the model $f$. The number of bins $B$ is a hyperparmeter of their method. For each bin $b \in [B]$, they construct an estimator $\hat{\pi}_b$ of $\pi_b = \mathbb{E}[f(X)|\mathcal{B}(X) = b]$ and its confidence interval , where $\mathcal{B} : \mathcal{X} \to [B]$ is the binning map. We used $\hat{f}(X) := \hat{\pi}_{\mathcal{B}(X)}$ as the prediction model for the performance evaluation.

## B.2    EXPERIMENT WITH BRADLEY-TERRY MODEL

Here we show the additional data generating process (DGP) following the Bradley-Terry (BT) model. Let $X1, X2 \sim \mathcal{N}(0, 1)$, $Z = r(X_1) - r(X_2)$ with $r(x) = 3\tanh(x)$, and $Y \sim \mathrm{Ber}(\sigma(Z))$ with $\sigma(x) = 1/(1 + e^{-x})$. The results were shown in Figure 4. As $\alpha$ increases, the proposed method attain the proper coverage of 90% even with a slightly better MSEs, which are similar to the results of the binary classification tasks shown in Section 6.1 and Figure 1. Figure 5 shows the distributions of the residuals $\hat{U}^j$ for $j = 1, 2$. One can observe that Assumptions 1 and 3 are satisfied, whereas Assumption 2 is not, from the figure.

# C    LLM EXPERIMENTS

## C.1    LCP FOR LLM PREFERENCE MODEL

We used the AlpacaFarm dataset (Dubois et al., 2023) (MIT License). We treated URM-LLaMa-3.1-8B (Lou et al., 2024) (OpenRAIL) as the gold reward model both to provide ground-truth preference labels for evaluation and to generate training preference labels via a BT model. The dataset labeled by URM-LLaMa-3.1-8B was split into $D^1$, $D^2$, $D^{\mathrm{cal}}$, and $D^{\mathrm{eval}}$ with 2400 samples each.

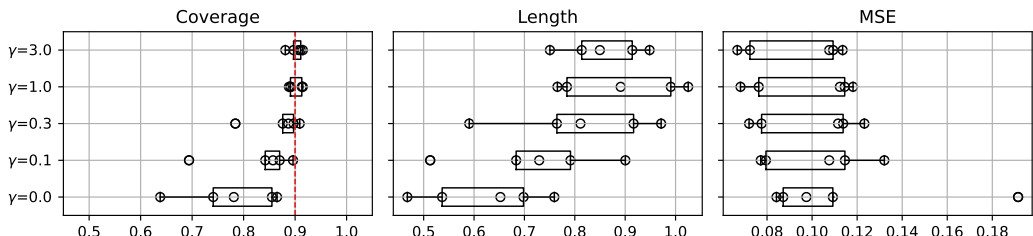

Figure 4: Performances of the proposed methods for the latent variable in the Bradley-Terry model.

As the target LLM-based reward model, we used `Skywork-Reward-Llama-3.1-8B-v0.2` (Liu et al., 2024) (Skywork AI License) and fine-tuned it with the loss equation 12. Each sample was represented by the 4096-dimensional final-layer output $X$ of the backbone LLM, serving as the input to the reward head. The reward head was a single linear layer (4096) with a dropout rate of 0.1. We trained with the Adam optimizer (learning rate $10^{-4}$, batch size 128) for 35 epochs.

EXAMPLES

Here are two examples from the Alpaca farm dataset with predicted probabilities $\Pr(Y_1 \succ Y_2|X)$ with 90% prediction intervals for prompts $X$ and outputs $Y_1$ and $Y_2$.

*Example 1*

> $X$: What is the gender of the French word ville?
> $Y_1$: The gender of the French word ville is feminine.
> $Y_2$: The French word ville is gender neutral. It does not have a gender in French.
> $\Pr(Y_1 \succ Y_2|X) = 0.772 \ [0.240, 0.973]$.

Although the predicted probability of preferring the first output over the second is 0.772, the 90% prediction interval $[0.240, 0.973]$ includes values below 0.5, indicating a chance that there is uncertainty about human preference in the opposite direction.

*Example 2*

> $X$: Summarize the benefits of a plant-based diet in one sentence.
> $Y_1$: A plant-based diet is beneficial for both the planet and our health, as it reduces meat, dairy and animal product consumption, decreasing environmental impact and risk of chronic diseases.
> $Y_2$: A plant-based diet provides numerous health benefits, including lowering risk of disease, providing essential nutrients, and being more sustainable.
> $\Pr(Y_1 \succ Y_2|X) = 0.937 \ [0.582, 0.994]$.

The output $Y_1$ clearly states how a plant-based diet is beneficial, while $Y_2$ is vaguer without specifying mechanisms or outcomes. This aligns with the human preference for $Y_1$ over $Y_2$, with 90% confidence, as indicated by the lower bound of the prediction interval $[0.582, 0.994]$ exceeding 0.5.

## C.2 Filtering Preference Data by Conformal Margins Improves LLM Alignment

We further evaluated the effect of filtering data using the prediction intervals obtained by LCP in the context of LLM alignment (RLHF), where an LLM is finetuned to human preference with a trained reward model. As a standard approach for LLM alignment, we adopted Direct Preference Optimization (DPO) (Rafailov et al., 2023), using the test split $D^{\mathrm{eval}}$ of our LLM dataset in Section 6.2 as the training dataset and applying the prediction intervals derived in Section 6.2 and 6.3. Specifically, we discarded preference pairs whose reward margins fell within the conformal prediction interval, as such pairs are less decisive in terms of preference ordering. This filtering aims to remove noisy or ambiguous preference pairs with small reward margins, which may otherwise hinder stable optimization in DPO by introducing label noise into the pairwise loss. By retaining only decisive pairs whose conformal intervals exclude zero, the training signal is expected to better reflect reliable human preferences.

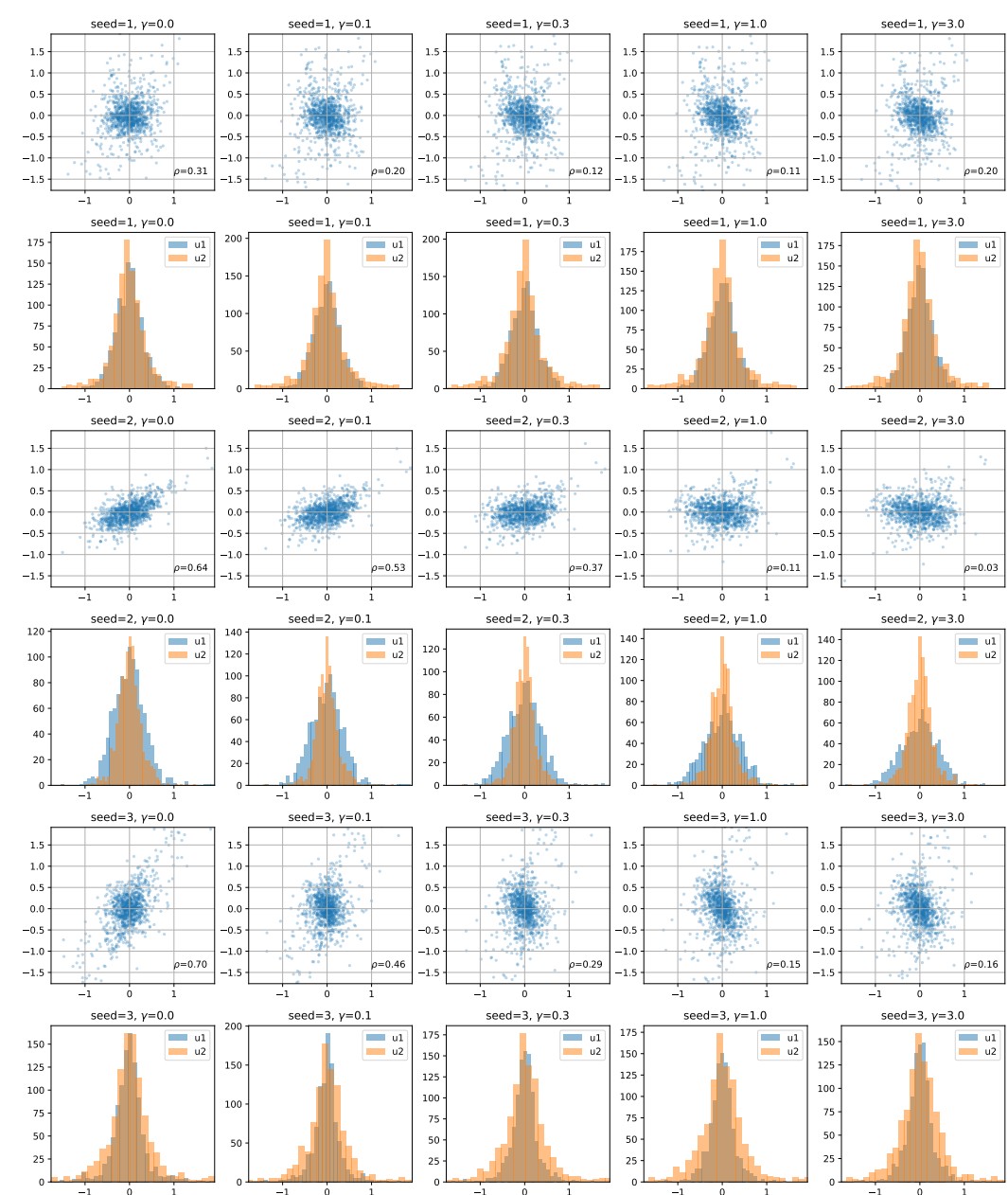

Figure 5: Distributions of residuals for the latent variable in the Bradley-Terry model according to different random seeds and orthogonalization strengths $\gamma$. The scatter plot of $\hat{U}^1$ (x-axis) and $\hat{U}^2$ (y-axis) visuzalied the joint distribution of the residuals with the correlation coefficients $\rho$. As the orthogonalization strength $\gamma$ increases, the correlation coefficients $\rho$ decreases. The historgram of the residuals visualized the symmetry of their distributions.

We compared three training dataset settings: `All` using all preference pairs, `LCP filtering` discarding pairs with margins inside the standard LCP interval, and `Localized LCP filtering` discarding pairs with margins inside the localized LCP interval described in Section 6.3. For reference, we also considered an `All_oracle` dataset in which labels were deterministically assigned by the gold reward model instead of being sampled probabilistically from the Bradley–Terry distribution.

We fine-tuned `Qwen2.5-1.5B-Instruct` with the three datasets described above. We evaluated the trained models with win rates on 512 prompts from the AlpacaFarm

`alpaca_human_preference` split, using the gold reward model `URM-LLaMa-3.1-8B` Lou et al. (2024). Table 1 reports the mean win rates and standard errors over 5 seeds. We found that conformal filtering improved alignment performance, with localized LCP filtering achieving the highest win rate and approaching the oracle setting. The hyperparameters, which were kept fixed across all settings, are summarized in Table 2.

| DPO dataset | Mean win rate | Std. error |
|---|---|---|
| All (without conformal filtering) | 0.5676 | 0.0090 |
| LCP filtering | 0.5744 | 0.0101 |
| Localized LCP filtering | **0.5834** | 0.0058 |
| All_oracle | 0.5951 | 0.0053 |

Table 1: DPO with/without conformal filtering. `All_oracle` uses deterministically assigned labels from the gold reward model instead of probabilistic sampling.

| Parameter | |
|---|---|
| Epochs | 2 |
| Learning rate | $1 \times 10^{-5}$ |
| $\beta$ | 0.1 |
| Batch size | 8 |
| LCP coverage rate | 0.2 |

Table 2: Experimental parameters of DPO experiment

## C.3 DISCUSSION

The LLM preference experiment illustrates a practical setting in which the latent variable of interest, the reward difference $Z = r(W, Y_1) - r(W, Y_0)$, is never observable in real-world data. Since modern RLHF pipelines rely on such latent rewards to steer large language models, understanding uncertainty in these reward values is crucial for assessing reliability and robustness.

Even when point estimates of reward differences are accurate, uncertainty around these estimates is highly consequential: (i) reward margins near zero cause unstable updates in preference-optimization methods (e.g., DPO); (ii) prediction intervals that cross zero naturally flag comparisons with ambiguous or noisy preference signals; and (iii) as shown in Section C.2, filtering out such ambiguous pairs improves RLHF stability and increases win rates. Hence, LCP provides a diagnostic tool for identifying preference uncertainty and improving downstream alignment performance.

Section C.1 includes examples where the point estimate $\tilde{f}(x)$ is moderately confident, yet the interval $C_{1-\alpha}(x)$ spans both positive and negative reward differences. These examples highlight how LCP captures uncertainty that is not visible from point predictions alone, providing more transparent preference signals.

In summary, the LLM experiments demonstrate that LCP applies to modern preference-learning settings, that coverage can be evaluated using oracle latent rewards, and that latent intervals have practical benefits for RLHF by identifying ambiguous preference pairs and stabilizing optimization.

## D PROOFS

*Proof of Proposition 1.* Computing the characteristic function of $\hat{V}$ as $\varphi_{\hat{V}} = \varphi_{\hat{U}_1 - \hat{U}_2} = \varphi_{\hat{U}} \varphi_{-\hat{U}} = (\varphi_{\hat{U}})^2$, we have equation 8. The first equality is by the definition of $\hat{V}$, the second is by Assumptions 1 and 2, and the third is by Assumption 3. $\square$

*Proof of Corollary 1.* Since the characteristics function of Gaussian distribution of a variance $\sigma^2$ is $\varphi(t) = \exp(\sigma^2 t^2/2)$, we have $\sqrt{\varphi(t)} = \exp(\sigma^2 t^2/4) = \exp[(\sigma/\sqrt{2})^2 t^2/2]$, which is the characteristics function of the Gaussian distribution of a variance $(\sigma/\sqrt{2})^2$. $\qquad\square$

*Proof of Proposition 2.* One can compute the characteristic function of $\tilde{U}$ as $\varphi_{\tilde{U}}(t) = \varphi_{(\hat{U}_1+\hat{U}_2)/2}(t) = \varphi_{\hat{U}_1+\hat{U}_2}(t/2) = \varphi_{\hat{U}^1}(t/2)\varphi_{\hat{U}^2}(t/2) = \varphi_{\hat{U}^1}(t/2)\varphi_{-\hat{U}^2}(t/2) = \varphi_{\hat{U}^1-\hat{U}^2}(t/2) = \varphi_{(\hat{U}^1-\hat{U}^2)/2}(t) = \varphi_{\hat{V}}(t)$, which implies $\tilde{U} \sim \hat{V}$. $\qquad\square$

*Proof of Theorem 1.* Since we assume the iid samples, the conformity score $\tilde{U}_i$ of $\tilde{f}$ for $i = 1, \ldots, n$ are iid, and thus, exchangeable. Let $\tilde{Q}_{1-\alpha}$ be the $(1-\alpha)(1+1/n)$-quantile of $|\tilde{U}_i|$ for $i$ in the calibration set. Then, by the standard argument of conformal prediction, $\tilde{C}_{1-\alpha}(X_{n+1}) = [\tilde{f}(X_{n+1}) - \tilde{Q}_{1-\alpha}, \tilde{f}(X_{n+1}) + \tilde{Q}_{1-\alpha}]$ guarantees the coverage $\Pr[Z_{n+1} \in \tilde{C}_{1-\alpha}(X_{n+1})] \geq 1 - \alpha$. By Proposition 2, $\tilde{U}$ and $V/2$ have the same distribution, hence the distribution of $(1-\alpha)(1+1/n)$-quantile of $|\tilde{U}_i|$ equals the distribution of $(1-\alpha)(1+1/n)$-quantile of $|V_i/2|$ for $i$ in the calibration set. Therefore, if we define $\hat{Q}_{1-\alpha}$ as the $(1-\alpha)(1+1/n)$-quantile of $|V_i/2|$ for $i$ in the calibration set and $\hat{C}_{1-\alpha}(X_{n+1}) = [\tilde{f}(X_{n+1}) - \hat{Q}_{1-\alpha}, \tilde{f}(X_{n+1}) + \hat{Q}_{1-\alpha}]$, then $\hat{C}_{1-\alpha}(X_{n+1}) = \tilde{C}_{1-\alpha}(X_{n+1})$ is guaranteed to cover the true $Z_{n+1}$ with probability at least $1 - \alpha$ after marginalizing over training and calibration sets. $\qquad\square$

*Proof of Theorem 2.* Let $k$ be a positive semi-definite kernel on $\mathbb{R}$. Suppose $k$ is bounded by $\kappa$ and 1-Lipschitz with a constant $L$:

$$|k(u+\delta, u'+\delta') - k(u,u')| \leq L(|\delta| + |\delta'|). \tag{17}$$

Let $\Delta := \hat{Z}^3 - Z$, then $U'^1 = U^1 + \Delta$ and $U'^2 = U^2 + \Delta$. The HSIC between two random variables $U$ and $V$ is given by:

$$\begin{aligned}\mathrm{HSIC}(U,V) = \mathbb{E}[k(U,U^*)k(V,V^*)] &+ \mathbb{E}[k(U,U^*)]\mathbb{E}[k(V,V^*)] \\ &- 2\mathbb{E}[\mathbb{E}[k(U,U^*)|U]\mathbb{E}[k(V,V^*)|V]],\end{aligned} \tag{18}$$

where $U^*$ and $V^*$ are i.i.d. copies of $U$ and $V$, respectively. We want to bound $|\mathrm{HSIC}(U'^1, U'^2) - \mathrm{HSIC}(U^1, U^2)|$. The difference of the first term of equation 18 is evaluated as follows:

$$\Delta^1 = \mathbb{E}[k(U^1+\Delta, U^{1*}+\Delta^*)k(U^2+\Delta, U^{2*}+\Delta^*) - k(U^1, U^{1*})k(U^2, U^{2*})]. \tag{19}$$

Using the equation $ab - a'b' = (a-a')b + a'(b-b') + (a-a')(b-b')$, this term can be bounded by

$$|\Delta^1| \leq 2\kappa L \mathbb{E}[|\Delta|] + L^2 \mathbb{E}[\Delta^2]. \tag{20}$$

The other two terms in equation 18 is bounded similary, yielding the following bound:

$$\mathrm{HSIC}(U^1, U^2) \lesssim \mathrm{HSIC}(U'^1, U'^2) + \mathbb{E}[|\Delta|] + \mathbb{E}[\Delta^2]. \tag{21}$$

$\qquad\square$

# E   USE OF LARGE LANGUAGE MODELS

LLMs were used for grammar and style suggestions.

