# OpenReview forum: "A Conformalized Inference on Unobservable Variables"
_ICLR.cc/2026/Conference — Submitted to ICLR 2026_

### Official Review · Reviewer_2v7e · 2025-10-26

**Soundness:** 2
**Presentation:** 2
**Contribution:** 3
**Rating:** 4
**Confidence:** 4

**Summary:**

This paper introduces Latent Conformal Prediction (LCP), a way to get coverage-valid prediction intervals for unobserved latent variables. The method splits data, trains two independent models, and uses their prediction discrepancy on a calibration split as a proxy for the residual distribution of an averaged predictor, enabling conformal calibration without ever observing the latent truth. To make this proxy identify the correct residual law, the authors posit mild assumptions on prediction residuals and propose residual orthogonalization. They demonstrate the approach on synthetic tasks and LLM-based preference learning, showing informative intervals and empirical coverage close to target levels, thereby extending CP to practical settings where the quantity of interest is never directly observed.

**Strengths:**

1. This work addresses a crucial but largely unexplored problem; the contribution is clear.
2. The proposed method is intuitive, lightweight, and creative.

**Weaknesses:**

1. The writing and presentation is unclear. Section 4 mixes problem setup with application examples; the general problem formulation should come first, with application-specific details moved to the experimental setup.
2. Framing the motivation through binary classification narrows the perceived scope. The paper’s true focus is uncertainty quantification for unobserved latent variables, and the narrative should emphasize this broader contribution.

**Questions:**

1. See weaknesses above.
2. The related work lacks discussion of CP methods aimed at improving prediction efficiency/informativeness in classification.
3. In Figure 1, what is α? If it denotes the miscoverage rate, it must lie in [0,1]; how can it be 3.0? Please clarify the axis/units or notation.
4. Please empirically validate the assumptions and provide a sensitivity analysis that characterizes performance as the degree of assumption violation increases.
5. Please provide intuition for the assumptions and for Theorem 2: What do the conditions mean operationally, and how do they translate into coverage?
6. How to adapt LCP to setting where data has potential distribution shifts?

---

> ### Author Response · Authors · 2025-12-01
>
> > The writing and presentation is unclear. Section 4 mixes problem setup with application examples; the general problem formulation should come first, with application-specific details moved to the experimental setup.
>
> We reorganized the structure of Section 4 by starting from Problem formulation, and then introducing its applications with new section headings. Please see the revised manuscript.
>
> > Framing the motivation through binary classification narrows the perceived scope. The paper’s true focus is uncertainty quantification for unobserved latent variables, and the narrative should emphasize this broader contribution.
>
> Thank you for the suggestion. We revised the paragraph (lines 57-62)  to clarify that binary classification is only a motivating example, and that our method targets the broader problem of constructing prediction intervals for general unobserved latent variables. The revised text now explicitly highlights this broader scope and includes the Bradley–Terry preference-learning example to make the generality clear.
>
> > The related work lacks discussion of CP methods aimed at improving prediction efficiency/informativeness in classification.
>
> Informativeness was discussed in a larger scope of the uncertainty quantification of predictions of latent variables in the introduction. For the efficiency, we understand the CP in general does not add any predictive/statistical efficiency of the model although our method seem to have efficiency gain by the residual orthogonalization.
>
> > In Figure 1, what is $\alpha$? If it denotes the miscoverage rate, it must lie in [0,1]; how can it be 3.0? Please clarify the axis/units or notation.
>
> Applogies for the confusing notation. The parameter $\alpha$ in Figure 1 should be $\gamma$, the strength of the orthogonal loss regularization in equation 13. We corrected Figure 1 in the revised manuscript.
>
> > Please empirically validate the assumptions and provide a sensitivity analysis that characterizes performance as the degree of assumption violation increases.
>
> We added the distributions of residuals from the experiment in the revised appendix, where one can observe that the orthogonal regularization actually reduces the dependence between residuals $\hat{U}^1$ and $\hat{U}^2$ which corresponds to the improvement in the coverage.
>
> > Please provide intuition for the assumptions and for Theorem 2: What do the conditions mean operationally, and how do they translate into coverage?
>
> The orthogonal loss equation 13 corresponds to the empirical mean of the first term of the right hand side of equation 18 with this first order kernel. Therefore, the optimization of equation 13 guarantees the optimization of the HSIC of the true residuals up to the error terms involving $\Delta$. We added this description in the revised manuscript.
>
> > How to adapt LCP to setting where data has potential distribution shifts?
>
> Thank you for the valuable question. One approach would be use the localized LCP described in the present paper to adapt the distribution shift given the data generating mechanism $P(Z|X)$ and $P(Y|Z,X)$ are the same in the original dataset. However, LCP under distribution shift in general should be the area of future research.

---

### Official Review · Reviewer_Teac · 2025-10-30

**Soundness:** 2
**Presentation:** 3
**Contribution:** 2
**Rating:** 2
**Confidence:** 3

**Summary:**

This paper proposes a new method called Latent Conformal Prediction (LCP) to quantify uncertainty and construct valid prediction intervals for unobservable latent variables.
Standard conformal prediction methods are unsuitable for this task as they require observed true labels, which are unavailable for latent variables. The core idea of LCP is to use sample-splitting to train two independent models; by analyzing the discrepancy between these two models' predictions on a calibration set, the method can estimate the distribution of the conformity score without ever seeing the true latent values.
To ensure the statistical identifiability of this residual distribution, the authors introduce a set of assumptions. The method's performance and ability to achieve guaranteed coverage rates are validated through experiments on synthetic binary classification data and a LLM preference learning task.

**Strengths:**

1. This paper provides a way to create valid prediction intervals for unobservable variables. This is important in the field of statistics.
2. The paper includes a mathematical proof (Theorem 1) that, under its assumptions, the prediction intervals it creates will achieve the desired coverage level.
3. This paper also proposes "Localized LCP", an extension that makes the prediction intervals adaptive.
4. The method is shown to work on both synthetic data and a real-world LLM preference learning task.

**Weaknesses:**

[Major Concern] I think the assumptions 1-3 are still too strong. It roughly means that one can construct independent (and symmetric) copies based on these assumptions. If so, with these assumptions, one could indeed construct the confidence interval. However, this hardly happens in reality.

Another thought: if those assumptions hold, it seems that one can directly use the interval [-M, M] where $M = quantile(|\hat{f}(X_i)|)$. Since $\hat{f}(X_i)$ is also independent copies (on the probability space over x and y given the calibration set), it seems that the coverage still holds here?

The authors try to solve this problem via the residual orthogonalization; however, it seems that this introduces another additional dependency on the trained parameter theta and the calibration fold. I think this is invalid in the conformal prediction literature.

Minor concern:
1. It would be better if the authors could conduct some more experiments on the hyperparameters gamma and K.
2. The data efficiency is a little bit loose here (not a big issue)
3. Seems that the coverage is not always guaranteed in the experiments? (fig1 and fig2)

**Questions:**

See above.

---

> ### Author Response · Authors · 2025-12-01
>
> > [Major Concern] I think the assumptions 1-3 are still too strong. It roughly means that one can construct independent (and symmetric) copies based on these assumptions. If so, with these assumptions, one could indeed construct the confidence interval. However, this hardly happens in reality.
>
> First, we realized, thanks to your feedback, that Assumption 2 (identity of residual distributions) is not needed for our main result (Theorem 1). We are grateful for your comment, which directly led us to discover and remove this assumption. (Please also refer to the response to the reviewer `AMij` for the Gaussian case and its extension.)
>
> Second, we now empirically verify Assumption 1 (orthogonality). Our proposed residual-orthogonalization algorithm effectively reduces the dependence between residuals $\hat U^1$ and $\hat U^2$. As shown in Figure 5 of the revised manuscript, their empirical correlation approaches zero after orthogonalization, and the coverage improves correspondingly. This provides practical evidence that the assumptions are attainable with modern neural predictors.
>
> Third, regarding the broader concern that Assumptions 1–3 resemble requiring “independent symmetric copies”: we show that these assumptions are far less restrictive in practice. For any continuously distributed random variable $V$ with CDF $F_V$, a monotone transform yields $U := F_V(V) \sim \mathrm{Unif}(0,1)$. That is, continuous random variables are always transformed to the uniform random variable with some monotonic transformation. It follows that when the binary outcome is generated by the process $1[f^\star(X) > Z^\star]$ with some continuous random variable $Z^\star$, then we can always write that $1[f(X) > Z]$ with $f = F_Z^\star(f^\star(X))$  and $F_Z^\star(Z^\star)$. This means the latent noise can always be represented in a standardized symmetric form without loss of generality, making it structural rather than restrictive.
>
> > Another thought: if those assumptions hold, it seems that one can directly use the interval [-M, M] where M=quentile(|f^(Xi)|). Since f^(Xi) is also independent copies (on the probability space over x and y given the calibration set), it seems that the coverage still holds here?
>
> Thank you for proposing a simple algorithm based on Assumptions 1- 3. For a new observation $X_{n+1}$, your proposal would output the prediction interval $C(X_{n+1})=[-M,M]$ with a guarantee $P(Z_{n+1}\in C(X_{n+1}))\ge 1-\alpha$? We respectfully do not think this interval makes sense because this prediction interval is constant and does not depend on the value of $X_{n+1}$.
>
> > The authors try to solve this problem via the residual orthogonalization; however, it seems that this introduces another additional dependency on the trained parameter theta and the calibration fold. I think this is invalid in the conformal prediction literature.
>
> By Theorem 2, the impact of information leakage through $\theta_3$ is upper bounded by the estimation error of $\theta_3$. When $D^{cal}$ is sufficiently large, $\theta_3$ accurately estimates the true latent variable $Z$, resulting in small values of $E[|\Delta|]$ and $E[\Delta^2]$. Therefore, minimizing $\mathrm{HSIC}(U'_1, U'_2)$ through regularization becomes approximately equivalent to achieving true independence $\mathrm{HSIC}(U_1, U_2)$. In this sense, the proposed method constructs asymptotically valid prediction intervals.
>
> > It would be better if the authors could conduct some more experiments on the hyperparameters gamma and K.
>
> Apologies for confusing you as $\alpha$ in Figure 1 is a typo for $\gamma$ as in Figure 2. The variable $K$ at line 297 is not the hyperparameter but the number of classes for a specific problem. We also note that Figure 1 in the revised manuscript already reports results for a range of $\gamma$ values, showing stable coverage across settings.
>
>
> > The data efficiency is a little bit loose here (not a big issue)
>
> As described at lines 324-325, the efficiency of the prediction model is recovered by ensembling the two prediction models $f^1$ and $f^2$. Empirically, Figures 2 and 4 show that the resulting MSE is close to that of the bootstrap method, which trains on the full dataset. This confirms that our ensemble approach effectively recovers the data efficiency in practice.
>
> > Seems that the coverage is not always guaranteed in the experiments? (fig1 and fig2)
>
> Because Assumptions 1-3 are approximately satisfied, the coverage is not strictly guaranteed. Furthermore, Algorithm 2 only orthogonalizes the first order correlations with a potential error term $\Delta$ displayed in Theorem2, which affects the coverage in practice. However, we empirically demonstrated Algorithm 2 drastically improved the coverage especially for the LLM experiment (Figure 2).
>
>
> Overall, your comments helped us substantially weaken the assumptions, verify them empirically, and better explain their generality. We sincerely appreciate this opportunity to improve the paper.

---

### Official Review · Reviewer_AMij · 2025-10-31

**Soundness:** 3
**Presentation:** 2
**Contribution:** 3
**Rating:** 6
**Confidence:** 2

**Summary:**

This paper proposed a way of constructing prediction intervals for unobservable latent variables. Though I'm not a subject matter expert, I still think the paper provides valid theoretical and empirical contributions to the field of Conformal Prediction.

**Strengths:**

1. It's a nice new problem formulation. I haven't seen this before. Studying latent variables intervals are a nice thing to have and I'm quite surprised the existing literature on this problem is so sparse.
2. The theoretical foundations are solid, and the coverage guarantees are nice. Specifically, I couldn't find any glaring issues with any of the proofs of the main claims.
3. Good exploration of the application of the technique. Covering all of LLM preferences to binary classification is very good to see for conformal prediction papers.
4. Addressing heteroskedascity is good to see. It's not a central claim in this paper to my understanding but it is nevertheless a good additon.
5. Nice addition with DPO. Again applying conformal prediction to help improve language modeling techniques is great!

**Weaknesses:**

None too serious.
1. Strong assumptions, but it's ok. Probably not possible to have results without them. Particularly, Assumption 2 is pretty strong, saying that the distribution of the residuals is equivalent. It would have been good to add an error analysis, if they are both guassians, based on the difference of the means.
2. The baseline mix could have been a bit stronger. A few more relevant baseliens would have been nice.  To my understanding, there are no real baselines of other methods in this paper? That would be really great to add.
3. Verifying the assumptions empirically would have been a nice addition. Especially assumptions 1 to 3 that assume properties on the distributions of the residuals. Perhaps you could take some datasets and measure this difference and see how reasonable these are?
4. The notation was a little hard to read IMO. Especially between propositions 1 and 2.

**Questions:**

1. Can you see if you can add an analsysis based on the difference of the measn of the distributions for Assumption 2?
2. Can you add more baselines to the paper? There aren't any right now.
3. Can you add an experiment to verify the assumptions?

---

> ### Author Response · Authors · 2025-12-01
>
> > The notation was a little hard to read IMO. Especially between propositions 1 and 2.
>
> Apologies for the confusing notation. We revised the notation, especially between propositions 1 and 2, where we discussed the identifiability of $\hat{U}$, not $\tilde{U}$.
>
> > Can you see if you can add an analsysis based on the difference of the measn of the distributions for Assumption 2?
>
> Suppose Assumptions 1 & 3 and $\hat{U}^1$ and $\hat{U}^2$ are gaussians: $\hat{U}^j\sim\mathcal{N}(0,\sigma_j^2)$. The distributions of the true residual $\hat{U}=(\hat{U}^1+\hat{U}^2)/2$, and our proxy $\hat{V}=(\hat{U}^1-\hat{U}^2)/2$ have the identical distribution $\mathcal{N}(0,(\sigma_1^2+\sigma_2^2)/4)$. In particular, in the Gaussian case we can drop Assumption 2, because the distribution of $U$ is already identified from that of $\hat{V}$ under Assumptions 1 and 3 alone.
>
> Motivated by your suggestion to analyze the effect of differences in the distributions under Assumption 2, we revisited our theory and realized that this relaxation is not limited to the Gaussian case. In the revised manuscript we prove that Proposition 2 and therefore Theorem 1 (the main coverage guarantee for LCP) continue to hold without Assumption 2: the distribution of the conformity score $\tilde{U}$ can be identified from $\hat{V}$ under Assumptions 1 and 3 only. We sincerely thank the reviewer for this insightful comment, which helped us simplify our assumptions and strengthen the main theorem.
>
>
> > Can you add more baselines to the paper? There aren't any right now.
>
> We thank the reviewer for pointing out the lack of baselines. In the revised manuscript, we added distribution-free binary classification intervals from Gupta et al. (2020) as a baseline method (shown in Figure 1). Their approach is the closest existing method that constructs uncertainty intervals for classification probabilities, and therefore represents the most relevant comparison point for our setting. Importantly, Gupta et al.’s method requires the latent variable to be representable as a conditional expectation of an observable quantity, which does not hold in our preference-learning setup (Section 4.2). This makes it an appropriate but challenging baseline. As shown in the updated Figure 1 (Coverage, Length, MSE panels), our method consistently achieves more reliable target coverage and shorter or comparable interval lengths compared with this baseline across all γ values. We also included the nonparametric bootstrap as an additional baseline to represent a widely used but non-coverage-guaranteed procedure. We have added the detail of the baseline in Section B.1.
>
>
>
> > Can you add an experiment to verify the assumptions?
>
> To address the reviewer’s request, we added an experiment that directly examines Assumptions 1–3 using the synthetic setting in Appendix. Figure 5 in the revised manuscript reports, for multiple seeds and regularization strengths $\gamma$, (i) scatter plots of $(\hat U^1, \hat U^2)$ together with the empirical correlation $\rho$, and (ii) overlaid marginal histograms of $\hat U^1$ and $\hat U^2$. This experiment confirms that orthogonal regularization systematically reduces the dependence between residuals (supporting Assumption 1): for example, the residual correlation decreases from ~0.6–0.7 at $\gamma = 0$ to values near zero as $\gamma$ increases.
>
> Importantly, we also observe that Assumption 2 (identical residual distributions) does not hold exactly; the histograms of $\hat U^1$ and $\hat U^2$ remain slightly different even for large $\gamma$. Nevertheless, this violation did not harm empirical coverage: the LCP intervals remain well calibrated across all settings, consistent with our theoretical result that Assumption 2 is not required for exact identifiability.
>
> Overall, this assumption-verification experiment demonstrates that (i) orthogonal regularization effectively enforces the independence condition, and (ii) even when Assumption 2 is violated in practice, coverage remains robust, matching our theoretical analysis (Theorem 1 and Theorem 2).

---

### Official Review · Reviewer_eqzF · 2025-11-04

**Soundness:** 3
**Presentation:** 3
**Contribution:** 2
**Rating:** 4
**Confidence:** 3

**Summary:**

This paper proposes a method called Latent conformal prediction to construct prediction intervals for latent variables instead of final outcomes. For example, the proposed approach offers a prediction interval of the predicted probability of classification instead of a confidence set of the class labels. The authors also use a regularization term to satisfy the orthogonality assumption required by their algorithm. Finally, experimental results on synthetic and real-world data are provided.

**Strengths:**

It is an easy-to-read paper. The paper is written in an intuitive approach. The authors showed an interesting use of conformal prediction for LLMs. The algorithm seemed very simple but effective.

**Weaknesses:**

Below I provide my comments.

### Major:
1. The authors mentioned the importance of obtaining correct latent variables in a system. However, it is not clear how a predictive interval of latent variables can help in decision-making. I would request the authors to make this precise.
2. Even though we do not have ${Z_i}_{i=1}^n$  in the given dataset,

can we extract
${Z_i}_{i=1}^n$

from the model trained on
${(X_i, Y_i)}_{i=1}^n$ ?

Then, a prediction interval can be constructed from the extracted ${Z_i}_{i=1}^n$. I would request the authors to clarify this.
3. It is not clear how the authors identify $\tilde{U}$ for $\tilde{f}$ by $\hat{V}/2$. The authors should provide some intuitive proof sketch in the main paper.
4. No baselines. The paper does not compare the proposed approach with existing conformal prediction or uncertainty quantification methods.
5. What’s the application of this approach? The authors should provide concrete examples or scenarios where constructing prediction intervals for latent variables is beneficial.

### Minor:
1. The discussion in the remark from lines 303 to 313 should be discussed more intuitively. The main point is slight obscure. It is also not clear how  characteristic function not having a compact support affects  estimating the distribution of $\tilde{U}$.  I would request the authors to make this clear.
2. If with Corollary 1, the authors do not need the Fourier transform to calculate the distribution of the conformity score. Then, what is the utility of the remark at line 303 discussing the Fourier transform and inverse Fourier transform?
3. The authors should provide more discussion on the LLM experiments.
4. It is not clear how the metrics, such as length and MSE, are used to evaluate the algorithm.
5. For calculating the MSE, how do the authors know the ground-truth latent variables in the LLM experiments?

**Questions:**

Here are my questions.

### Questions:
1. We have access to the outcomes $Y$ in our dataset. However, the latent variables should depend on the models we are training. How can that be model-agnostic? For example, the authors mentioned preference learning with the Bradley–Terry model. Since the reward depends on the model, the predictive interval will not be model-agnostic, right? I would request the authors to explain this.
2. What did the authors mean by saying, “Instead of the logit of the probability of $Y = 1$, the problem is not normalized because there is one-dimensional freedom for translations of the reward $r(W, Y)$”?
3. Why does Assumption 1 generally not hold?
4. “We could expect the symmetric fluctuation of the predictions with respect to the random perturbation of the observation in the training set.” How do the authors randomly perturb the observations?

---

> ### Author Response · Authors · 2025-12-01
>
> > The authors mentioned the importance of obtaining correct latent variables in a system. However, it is not clear how a predictive interval of latent variables can help in decision-making. I would request the authors to make this precise.
>
> In addition to the applications in LLMs, the predictive interval plays a prominent role in high-stakes decision making. In the medical domain, where physicians are required to make high-stakes decisions based on imperfect real-world data, the importance of confidence intervals in diagnostic inference is well-established. The STARD 2015 guideline explicitly requires that diagnostic accuracy metrics be reported with 95% CIs to convey the precision of estimates (Bossuyt et al., JAMA 2015). In clinical reasoning, uncertainty around post-test probabilities guides whether to treat or to seek additional testing (Pauker & Kassirer, NEJM 1980). Moreover, recent discussions in medical AI emphasize that communicating uncertainty is critical for safety and trust: Begoli et al. (Nat Mach Intell 2019) highlight that uncertainty quantification is a prerequisite for reliable clinical AI systems, and Kompa et al. (NPJ Digit Med 2021) show that expressing confidence or prediction intervals helps clinicians decide when to act or defer. Therefore, quantifying uncertainty via confidence or prediction intervals is essential for risk-aware diagnostic and treatment decisions, directly motivating our latent-variable conformal framework.
>
>
> > Even though we do not have $Z_i$ in the given dataset, can we extract $Z_i$ from the model trained on $(X_i, Y_i)$ ?Then, a prediction interval can be constructed from the extracted Zi. I would request the authors to clarify this.
>
> Yes, $(Z_i:i\in D^j)$ are predicted by the model $\hat{f}^j$ learned from dataset $(X_i, Y_i:i\in D^j)$ for $j=1,2$. Based on the predicted $(Z_i:i \in D^1)$ and $(Z_i:i \in D^2)$ one can construct a prediction interval for the model $\tilde{f}=(\hat{f}^1+\hat{f}^2)/2$.
>
>
> > It is not clear how the authors identify $\tilde{U}$ for $\tilde{f}$  by $\hat{V}$. The authors should provide some intuitive proof sketch in the main paper.
>
> Assumption 1 and 3 directly yields the equivalence $\hat{U}^1 + \hat{U}^2 \sim \hat{U}^1 - \hat{U}^2$. We added this in Remark right after Theorem 2.
>
>
> > No baselines. The paper does not compare the proposed approach with existing conformal prediction or uncertainty quantification methods.
>
> We added a baseline method in the revised manuscript, specifically the distribution-free confidence-interval approach of Gupta et al. (2024), which is the closest existing method applicable to our problem setting. As shown in Figure 1, our method consistently outperforms this baseline.
>
>
> > What’s the application of this approach? The authors should provide concrete examples or scenarios where constructing prediction intervals for latent variables is beneficial
>
> We believe this question is related to question 1 above in high stakes decision making. Another application is RLHF which we demonstrate its value in the appendix.
>
> > The discussion in the remark from lines 303 to 313 should be discussed more intuitively. The main point is slight obscure. It is also not clear how characteristic function not having a compact support affects estimating the distribution of U~. I would request the authors to make this clear.
>
> We will include the following discussion in our revision. Since the empirical characteristics function $\hat{\varphi}_n(t)=n^{-1}\sum_ie^{iX_it}$ of samples $(X_i:i\in[n])$ does not have compact support, the estimated characteristic function $\varphi_U(t)=\sqrt{\hat{\varphi}_n(t)}$ does not as well. Distribution function $\hat{F}(u)$ is obtained by the inverse Fourier transform $\hat{F}(u) = \int \hat{\varphi}_n(t) e^{-itu}dt$, where the integration is taken across $[-\infty,\infty]$. However this diverges unless we limit the window for the integration in some finite interval $[-T,T]$.
>
> > If with Corollary 1, the authors do not need the Fourier transform to calculate the distribution of the conformity score. Then, what is the utility of the remark at line 303 discussing the Fourier transform and inverse Fourier transform?
>
> Corollary 1 is only valid for Gaussian $\hat{U}$. However, for general $\hat{U}$, we need the Fourier transforms when applying the Kotlarski’s lemma (Proposition 1). In this case, the difficulty in numerical computation arises as described in Remark.
>
>
> > The authors should provide more discussion on the LLM experiments.
>
> We thank the reviewer for pointing out the need for additional discussion of the LLM experiments. We have now added a dedicated subsection (Appendix, Discussion) that clarifies the motivation, interpretation, and practical relevance of the LLM preference-ranking experiments.

---

> > ### Author Response · Authors · 2025-12-01
> >
> > > It is not clear how the metrics, such as length and MSE, are used to evaluate the algorithm.
> >
> > We should have explained the rationale to evaluate the length of prediction interval and MSE of the proposed algorithm. We evaluate the length alongside the coverage to make sure that the proposed algorithm does not achieve the coverage by stretching the prediction interval too wide. The evaluation of MSE provides the evidence that the proposed sample-split approach does not sacrifice the model accuracy by comparing with the bootstrap.
> >
> >
> > > For calculating the MSE, how do the authors know the ground-truth latent variables in the LLM experiments?
> >
> > The experiments are based on synthetic and semi-synthetic data where we have access to the ground-truth latent variables. Note that our LLM experiment is also semi-synthetic. Following common practice in RLHF studies (e.g., Gao et al., 2023; Caste, 2024), true human latent rewards are not observable, so we use a high-quality reward model (“gold RM”) as a proxy for the ground truth. Once the gold RM is fixed, the Bradley–Terry model uniquely determines the latent reward difference
> > $Z^\star = r^\star(X,Y_1) - r^\star(X,Y_0)$,
> > which we use as the ground-truth latent variable when computing MSE.
> > * Leo Gao, John Schulman, Jacob Hilton, “Scaling Laws for Reward Model Overoptimization”, ICML 2023.
> > * Thomas Coste, Usman Anwar, Robert Kirk, David Krueger, “Reward Model Ensembles Help Mitigate Overoptimization”, ICLR 2024.
> > * Leo Gao, John Schulman, Jacob Hilton, “Scaling Laws for Reward Model Overoptimization”, ICML 2023.
> > * Thomas Coste, Usman Anwar, Robert Kirk, David Krueger, “Reward Model Ensembles Help Mitigate Overoptimization”, ICLR 2024.
> >
> > > We have access to the outcomes  in our dataset. However, the latent variables should depend on the models we are training. How can that be model-agnostic? For example, the authors mentioned preference learning with the Bradley–Terry model. Since the reward depends on the model, the predictive interval will not be model-agnostic, right? I would request the authors to explain this.
> >
> > Bradeley-Terry model is the model about the structure, or data generating process, of the variables, but does not refer to any specific machine learning model. For example, we can use any machine learning models to fit the reward function $r(W,Y)$. The proposed method works with any of these models, so are model-agnostic.
> >
> >
> > > What did the authors mean by saying, “Instead of the logit of the probability of Y=1, the problem is not normalized because there is one-dimensional freedom for translations of the reward r(W,Y)”?
> >
> > In the logistic regression for binary classification tasks, we only model the difference of the rewards $r(W,1)-r(W,0)$. However, if we want to model both rewards $r(W,0)$ and $r(W,1)$, we have one-dimensional freedom since $r(W,0)+\delta$ and $r(W,1)+\delta$ give the same probability for any scalar $\delta$.
> >
> >
> > > Why does Assumption 1 generally not hold?
> >
> > Because both models are trained on data drawn from the same distribution (and often share architecture), their residuals are generally correlated. Assumption 1 is approximated by random resplitting or by our proposed orthogonal regularization.
> >
> >
> > > “We could expect the symmetric fluctuation of the predictions with respect to the random perturbation of the observation in the training set.” How do the authors randomly perturb the observations?
> >
> > We do not add artificial noise to the data. The “random perturbation” refers to the natural randomness coming from (i) the random split into $D^1$ and $D^2$, and (ii) stochastic training (random initialization, minibatch sampling, dropout). Because each model is trained on a randomly perturbed version of the dataset, their prediction residuals fluctuate symmetrically around zero, which motivates Assumption 3.

---

### Author Response · Authors · 2025-12-01

We thank all reviewers for their thoughtful and constructive comments. Your feedback significantly helped us improve both the clarity and the theoretical precision of the manuscript. In particular, after revisiting the identifiability argument, we have significantly strengthened the main theoretical contribution by proving Proposition 2 and Theorem 1 under substantially weaker conditions: specifically, we establish our main results without any reliance on Assumption 2. The revised proofs are now independent of this assumption, and a new experimental evaluation (Figure 5) confirms the robustness of the findings in its absence. We also clarified several conceptual points raised by reviewers and strengthened the exposition in the main text. We are grateful for the opportunity to refine our work.

---

### Meta-Review · Area_Chair_U9si · 2026-01-04

**Summary:**

This paper addresses the novel and important problem of constructing prediction intervals for unobservable latent variables. Reviewers agree that the problem formulation is original, the approach is intuitive, and the theoretical analysis is nontrivial. The application to LLM preference learning and DPO is also viewed positively.

However, the paper relies on very strong assumptions regarding residual independence and symmetry, whose realism and necessity are insufficiently justified. As noted by reviewers, under these assumptions, simpler interval constructions may already achieve coverage, raising concerns about the necessity of the proposed method. Moreover, the residual orthogonalization step may introduce dependencies that challenge conformal validity, and these issues are not fully resolved.

Empirically, the evaluation lacks strong baselines, limited sensitivity analysis, and does not adequately examine robustness to assumption violations. Some coverage deviations are also observed. Finally, clarity and organization issues (notation, problem framing, and section structure) further weaken the presentation.

Overall, despite promising ideas, the theoretical and empirical concerns prevent acceptance at this time. A substantial revision addressing assumption realism, methodological necessity, empirical robustness, and exposition would benefit the quality during revising the paper.

**Reviewer Concerns:**

- Strong assumptions by several reviewers are still outstanding.
- Lack of baselines are still outstanding.
- Lack of practice of how interval-based prediction benefit the decision-making, especially in the era of LLMs, are still outstanding.

**Reviewer Scores:**

After reading the rebuttal, I think that no reviewer will change their score.

---

### Decision · Program_Chairs · 2026-01-26

Reject